# Cumulative learning enables convolutional neural network representations for small mass spectrometry data classification

Khawla Seddiki[1,2], Philippe Saudemont[2], Frédéric Precioso[3], Nina Ogrinc [2], Maxence Wisztorski [2], Michel Salzet[2], Isabelle Fournier [2✉] & Arnaud Droit [1✉]

Rapid and accurate clinical diagnosis remains challenging. A component of diagnosis tool development is the design of effective classification models with Mass spectrometry (MS) data. Some Machine Learning approaches have been investigated but these models require time-consuming preprocessing steps to remove artifacts, making them unsuitable for rapid analysis. Convolutional Neural Networks (CNNs) have been found to perform well under such circumstances since they can learn representations from raw data. However, their effectiveness decreases when the number of available training samples is small, which is a common situation in medicine. In this work, we investigate transfer learning on 1D-CNNs, then we develop a cumulative learning method when transfer learning is not powerful enough. We propose to train the same model through several classification tasks over various small datasets to accumulate knowledge in the resulting representation. By using rat brain as the initial training dataset, a cumulative learning approach can have a classification accuracy exceeding 98% for 1D clinical MS-data. We show the use of cumulative learning using datasets generated in different biological contexts, on different organisms, and acquired by different instruments. Here we show a promising strategy for improving MS data classification accuracy when only small numbers of samples are available.

[1] Computational Biology Laboratory, CHU de Québec - Université Laval Research Center, Québec City, Québec, Canada., Québec City, QC, Canada. [2] Université de Lille, Inserm, CHU Lille, U1192-Protéomique Réponse Inflammatoire Spectrométrie de Masse-PRISM, Lille F-59000, France. [3] Université Côte d'Azur, CNRS, INRIA, I3S, Sophia Antipolis, France. ✉email: isabelle.fournier@univ-lille.fr; arnaud.droit@crchudequebec.ulaval.ca

Accurate and rapid identification of cancer tissues has a crucial impact on medical decisions. Conventional histopathological examinations are resource intensive and time-consuming, requiring 30–45 min per sample to be processed and the presence of skilled pathologists[1]. A similar need exists in the treatment of infections, where accurate identification of microorganisms responsible for human infection is important to ensure the most appropriate and effective treatment for a patient, in the shortest possible time[2]. In this context, it is essential to use methods which provide accurate identification of the analyzed samples. Mass spectrometry (MS) is particularly useful for such purposes since it provides non-targeted molecular information on the millisecond time scales. Its sensitivity, reproducibility, and suitability for analyzing complex mixtures are well established. New analysis methods of crude samples are making diagnosis even faster and easier. Simultaneously, the development of MS-based bacterial biotyping illustrates the value of MS in rapid clinical applications[3].

For cancer-related diagnosis and microbial pathogen identifications, many popular classification machine learning (ML) models, such as support vector machine (SVM)[4], random forest (RF)[5], and linear discriminant analysis (LDA)[6] have been already used and compared[7–10]. These ML methods are applied to pre-processed MS data, but differences in preprocessing pose a major challenge to any comparison analysis. Classification model design for rapid applications thus becomes a highly complex task, since it must follow a workflow involving several interdependent pre-processing steps. Data preprocessing is used to improve the robustness of subsequent multivariate analysis and to increase data interpretability by correcting issues associated with MS signal acquisition[11]. Preprocessing quality is important, and if inadequate, can lead to biased or biologically irrelevant conclusions[12]. Several factors, often related to the experimental conditions including sample heterogeneity, sample processing, MS analysis (e.g. electronic noise, instrument calibration stability, temperature stability, etc.), and other experimental conditions can contribute to spectral variations including shifts in peak location, fluctuating intensities, and signal distortion[13]. In other words, peaks corresponding to the same molecule in different samples can be shifted and their signal intensity can vary from one spectrum to another[14,15]. Signals of lower intensity are in general more affected by such variations because they can become buried in the baseline noise in certain cases. Since these include many markers of interest, this may lead to a loss of important biological information[16]. Corrections on peak position variations are required in order to align different spectra properly and thus ensure consistency in downstream analysis. This alignment constitutes a significant hindrance to achieving reproducibility especially in today's complex datasets, and remains a challenging problem since it is neither linear nor uniform across the whole collection of MS spectra[16]. In addition to peak shifts, other spectral fluctuations must be corrected in order to minimize background, serious intensity distortion due to noise and baseline drift caused by instrument electronics, ion saturation, or contaminants within the samples[13]. To overcome batch effects, peak intensities must be equalized to reduce overall signal variation between acquisitions using intensity calibration or normalization[17,18]. Log-intensity transformation is one of the methods most commonly used to attenuate large differences in variability differences between peaks across the spectrum[18]. Another preprocessing step that is crucial to subsequent analysis is the peak detection, also known as peak picking. This consists of identifying informative peaks that correspond to a true biological signal by finding all local maxima in the spectrum, it corresponds to the conversion of spectra from a profile to a centroid mode[19]. Finally, the curse of dimensionality, must be avoided. This is a well-known problem that arises when processing MS data having a large number of dimensions, and is lessened using data dimensionality reduction techniques to remove irrelevant or redundant features[20]. Various MS classification workflows have been developed so far, but there are no golden standards for the optimal choice of parameters at each individual step, for their quality evaluation or for their best combination[21]. It has been shown that the choice of preprocessing parameters for a specific dataset can decrease the performance of the classification model and that preprocessing may be effective only for that dataset and not for any new others generated from different instruments or with different settings[22]. A standard pipeline for MS classification using SVM, RF, or LDA must include these preprocessing steps and must consider aforementioned constraints, which makes such algorithms unsuitable for rapid analysis.

Convolutional neural networks (CNNs) are one of the most successful deep learning architectures designed to learn representation from an input signal with different levels of abstraction[23]. A typical CNN includes convolutional layers which learn spatially invariant features from input (i.e. invariance to translation, invariance to scale, etc.) stored in feature maps, pooling operators that extract the most prominent structures, and fully connected layers for classification[24]. To address rapid clinical MS data classification tasks, CNNs represent an attractive approach offering various advantages over conventional ML algorithms. These include significantly higher accuracy, effectiveness on raw spectra even in presence of signal artifacts and hence discards the need for data preprocessing before classification[25]. Besides the integration of features extraction with classification and without a feature-engineering step since all layers are trained together. Finally, the exploitation of spatially stable local correlations by enforcing the local connectivity patterns, where the output of each layer of these networks is directly related to small regions of the input spectrum. However, CNNs classification efficiency trained using a small number of spectra drops rapidly[25]. Unfortunately, many real-world applications do not have access to big training sets because of data scarcity, or because of the difficulty and expense in labeling data[26]. In medicine, it is often the case that some samples are only accessible in limited amounts, especially for rare diseases (e.g. biopsies). Therefore the size of clinical datasets is constrained by data availability and by the experiments complexity and high cost[27]. For such applications, transfer learning has emerged as an interesting approach[28]. This technique is applicable to small datasets and therefore requires fewer computational resources while increasing the classification accuracy as compared to CNNs models built from scratch. Transfer learning is a two-step process. An accurate data representation is first learned by training a model on a dataset containing a large amount of annotated data covering many categories. This representation (i.e. its model weights) is then reused to build a new model based on a smaller annotated dataset containing fewer categories, by training only the final decision layer(s) or by also fine-tuning the whole model with the reduced set of categories. Transfer learning has proven useful in many engineering areas including computer vision, robotics, image classification, and natural language processing (NLP) applications[29]. Adapting this principle to MS data, basic similarities between spectral profiles gathered from different datasets would be used to address new classification problems. Most of MS classification works based on CNNs focus on MS 2D-imaging analysis[30,31]. Thus Transfer learning has yet to be explored for 1D spectral data, since no 1D spectral dataset as large as the largest 2D image dataset, ImageNet[32], is available. Only few studies of input signal classification or regression using 1D-CNNs with vibrational spectroscopy data[33], near-infrared (NIR) spectroscopy data[34–36] or Raman spectroscopy data[25] have been published. As far as we know,

**Table 1 Overall accuracies of SpiderMass spectra classification using three CNN architectures.**

| Datasets | # classes | variant_Lecun | variant_LeNet | variant_VGG9 |
|---|---|---|---|---|
| Canine sarcoma | 2 | **0.98 ± 0.00** | 0.96 ± 0.01 | 0.96 ± 0.01 |
| | 12 | 0.88 ± 0.03 | 0.88 ± 0.02 | **0.90 ± 0.01** |
| Microorganisms | 3 | **0.91 ± 0.03** | 0.52 ± 0.11 | 0.67 ± 0.09 |
| | 5 | **0.89 ± 0.02** | 0.68 ± 0.03 | 0.61 ± 0.13 |

The best result for each task (accuracy ± standard variation over 10 independent iterations) is indicated in boldface.

none of these works have led to transfer learning approaches on 1D-MS data.

The aim of this study is to challenge CNN models for classification tasks of 1D mass spectra when the training set is very small, to evaluate the weaknesses of transfer learning in such a context, and finally to design an approach: cumulative learning. Pattern recognition models are built using small clinical datasets generated for the diagnosis of cancers or infections.

Our proposed approach differs from standard transfer learning by different aspects:

- The number of output classes: Because of its abundant categories and large number of images, ImageNet is used widely as the source dataset in transfer learning cases. The typical transfer learning operation consists of using a pre-trained model, for instance on 1000 different ImageNet dataset classes and applying it to a new classification problem (possibly after fine-tuning to adapt to the new problem), which usually involves a much smaller number of classes to be predicted. In this study, the transfer learning approach comprised training a model on a dataset with only two output categories and efficient transfer of this model to classification problems with 2, 3, 5, and even 12 output categories.

- The diversity of the target tasks: In standard transfer learning, the target tasks are similar and thus rely on similar input data features (i.e. image classification task). In this study, transfer and cumulative learning are applied to different biological contexts (i.e. diseases) that are unlikely to share common features. Our results show that CNNs are powerful tools for learning "potentially generic" representations from spectra having no intuitive relationship to the medical target or sensitivity to acquisition instrument diversity.

- The accumulation of learning representations through several phases of representation model training up to the final decision level (fully connected layers and softmax/sigmoid layer): Standard transfer approaches to learn generic representations require the initial model to be trained with as much data as possible integrating all the essential features from the input data. This leads to poor results with small datasets. In this study, instead of considering only transfer learning (which would be a one-shot representation model), the same representation model is trained for several tasks successively to converge to an optimal model. It thus learns cross-classification tasks and cross-instrument representations and thereby becomes capable of smoothing fluctuations in instruments performance, which leads to a significant improvement in classification accuracy.

This work details our CNNs framework and its application on small clinical datasets classification. The approach is fully applicable to other domains where the lack of training data is still a hindrance.

## Results

**CNN architectures performances**. Three CNN architectures are compared using cancer and microorganism datasets. In order to

evaluate the effect of varying the number of CNN layers on classification performance, CNNs containing four (variant_Lecun, model 1), five (variant_LeNet, model 2), or nine layers (variant_VGG9, model 3) are evaluated and compared (Supplementary Fig. 3). The statistical significance in classification accuracies between the first and the second best results is computed with a $t$-test over 10 independent iterations ($p$-values < 0.001). For canine sarcoma classification, binary (2 classes) classification of tissues as healthy or cancerous is sought first, followed by differentiation of sarcoma types (12 classes). For the microorganism dataset, two multi-class classifications based on standard classification for clinical purposes are considered. The first is a 3-class model intended to identify the sample as yeast, Gram-positive bacteria, or Gram-negative bacteria. The second model is intended to allow identification of each of the five microorganisms. Table 1 lists the classification accuracy for each dataset. All sensitivities, specificities, and confusion matrix metrics associated with each dataset are described in the Supplementary Tables 3–10.

All three CNNs architectures perform badly on 3 of the 4 tasks, which is not surprising because of the low number of spectra used for the training. Variant_Lecun is the best at binary classification of canine sarcoma, but when the number of classes is expanded to 12, variant_VGG9 is slightly better. Errors in the confusion matrix are distributed uniformly across classes (Supplementary Table 6). Variant_Lecun is the best at classifying microorganisms, using the 3-class or the 5-class model. Accuracies suffer quickly from over-fitting when a deep architecture such as variant_LeNet and variant_VGG9 are used on data of this size. The only classification that could be described as accurate is for canine sarcoma versus healthy tissue (binary classification) by variant_Lecun with an average accuracy of 0.98. Based on this result, we focus our subsequent efforts on the canine sarcoma and microorganism multi-class classifications.

**Transfer learning performances**. In order to improve the classification performance, we use CNN architectures trained on the large MALDI-MSI rat brain dataset and test them on small clinical datasets. We obtain nearly (0.99 ± 0.00) for the rat brain dataset binary classification with the three CNN architectures. Transfer learning allows the model to learn and detect generic representations of MS peaks. By freezing the lower CNN levels, we assume that the model extracts the right patterns from the MALDI-MSI spectra, and that only the high level is needed to take into account specific SpiderMass peak features. As shown in Table 2, transfer learning clearly improves the classification accuracies of both small SpiderMass datasets compared to CNN models trained from scratch (without transfer learning).

Gains in the accuracy of canine sarcoma differentiation are somewhat obtained for all three architectures, although improvements are still needed. Variant_LeNet and variant_VGG9 predict the correct classes with almost equal success, but both fail to separate some classes, as shown in the confusion matrix in Supplementary Table 12. The 3-class microorganisms classification is improved for all three architectures. Improvements are

**Table 2 Overall accuracies of SpiderMass spectra classification using three CNN architectures after transfer learning.**

| Datasets | # classes | variant_Lecun | variant_LeNet | variant_VGG9 |
|---|---|---|---|---|
| Canine sarcoma | 12 | 0.90 ± 0.01 (02%) | **0.92 ± 0.01** (04%) | **0.93 ± 0.02** (03%) |
| Microorganisms | 3 | **0.99 ± 0.00** (08%) | 0.96 ± 0.01 (84%) | 0.95 ± 0.02 (41%) |
|  | 5 | **0.99 ± 0.00** (11%) | **0.99 ± 0.00** (45%) | 0.96 ± 0.02 (57%) |

The best result for each task (accuracy ± standard variation over 10 independent iterations) is indicated in boldface. The improvement in performance from scratch is expressed as a percentage.

**Table 3 Overall accuracies of canine sarcoma spectra classification by the three CNN architectures.**

| Protocol | variant_Lecun | variant_LeNet | variant_VGG9 |
|---|---|---|---|
| Scenario A | 0.92 ± 0.01 (04%[a] 02%[b]) | **0.95 ± 0.01** (08%[a] 03%[b]) | 0.94 ± 0.01 (04%[a] 01%[b]) |
| Scenario B | 0.95 ± 0.02 (08%[a] 05%[b] 03%[c]) | **0.99 ± 0.00** (12%[a] 07%[b] 04%[c]) | 0.96 ± 0.00 (06%[a] 03%[b] 02%[c]) |

The best result for each task (accuracy ± standard variation over 10 independent iterations) is indicated in boldface.
[a]The improvement is expressed as a percentage relative to learning from scratch.
[b]The improvement is expressed as a percentage relative to transfer learning.
[c]The improvement is expressed as a percentage relative to Scenario A.

considerable also for the 5-class task. Transfer learning by variant_Lecun leads to the best performances in the experiment. These results suggest that training a CNN model with extracted spectral features transferred even from an unrelated field is better than training it with spectral features learned from scratch with a small dataset. The aim of the following experiments is to improve the canine sarcoma multi-class classification performance.

**Cumulative learning performances**. To improve further the accuracy of the canine sarcoma multi-classification, CNNs are trained using the large MALDI-MSI rat brain dataset and then fine-tuned using the SpiderMass datasets. Two scenarios are tested: (Scenario A) training on intermediate beef liver and then canine sarcoma dataset. We obtain nearly (0.98 ± 0.00) for beef liver dataset binary classification with the three CNN architectures. Although not biologically related to the sarcoma context, beef liver recognition allows the model to appropriate the clinical data and their specific characteristics to improve its generalization capability in the second step; (Scenario B) training on beef liver, then on microorganisms and lastly on canine sarcoma dataset.

As shown in Table 3, Scenario A improves the classification accuracies considerably relative to learning from scratch and slightly relative to transfer learning, the best improvement is obtained for variant_LeNet. Scenario B provides a slight additional improvement over Scenario A, and the greatest accuracy is achieved also with variant_LeNet architecture. The effectiveness of the cumulative knowledge method is thus apparent, enabling the CNNs to distinguish not only cancerous versus healthy tissues (binary classification), but also the different cancer types (see confusion matrices in Supplementary Table 18 for Scenario A and in Supplementary Table 20 for Scenario B) despite the large number of classes, the small size, and the heterogeneity of the dataset.

We test CNNs configured with different numbers of frozen layers using transfer and cumulative learning in order to evaluate the trade-off between freezing and fine-tuning. Freezing all convolutional layers (i.e. the representation portion) and re-training all fully connected layers (i.e. the decisional portion) gives a configuration that outperforms the others. Except for Scenario B where the best architecture is obtained by freezing all convolutional layers barring the last one. We test the same protocols on datasets with a smallest bin (binned at 1 instead of 0.1, see Supplementary Tables 50–52). Similar improvements in accuracy are observed, except that the variant_VGG9 architecture

outperforms other networks on the canine sarcoma dataset. This may suggest that an architecture with three convolutional layers performs well with data binned at 0.1 (15,000 features), while a deep architecture such as variant_VGG9 performs well in case of a more compressed data (1500 features).

Cumulative learning strategy brings new questions: how generalizable is the final representation after several steps of cumulative learning? Is the final representation more specifically adapted to the last dataset used to accumulate MS knowledge? Let us first remind that the classification accuracy obtained by CNNs from scratch on data used for training (rat brain and beef liver) and after transfer learning for microorganisms (Table 2) is between 0.98 and 0.99. Testing the final cumulative representation of variant_LeNet (purple arrows from Scenario B in Fig. 1) on rat brain, beef liver, and microorganism datasets separately preserved a classification accuracy between 0.98 and 0.99. This indicates that the CNN model accumulates MS knowledge through the successive training phases without any loss of generalization. It suggests that a "generic" representation of MS data for classification tasks might exist and that the resulting cumulative representation is robust to the organism, to the tissue phenotype, and to the instrument variability.

**External public datasets results**. The classification accuracy on the two human ovary datasets were compared previously to explain how the choice of the MS instrument, its resolution or preprocessing steps become an obstacle to the reproducibility and reliability of pattern recognition. The aim of this experiment is not to compare our results to the reported results in the original paper analyzing these datasets, because the use of a commercial software for the classification, and then the preprocessing strategy is different in the original paper[37]. Our purpose is to demonstrate the efficiency of our learning methodology capable of handling multiple MS features: ionization sources (from WALDI to SELDI), resolutions (from high to low), and mass ranges (from lipids to proteins). We assess CNNs performance using the same training and evaluation approach, only with variant_LeNet architecture because of its superior performance with SpiderMass datasets and its low computational resources needed. Variant_LeNet is thus trained on the rat brain dataset as the source dataset, followed by the transfer learning protocol using the high-resolution dataset and cumulative learning Scenario A using the low-resolution dataset.

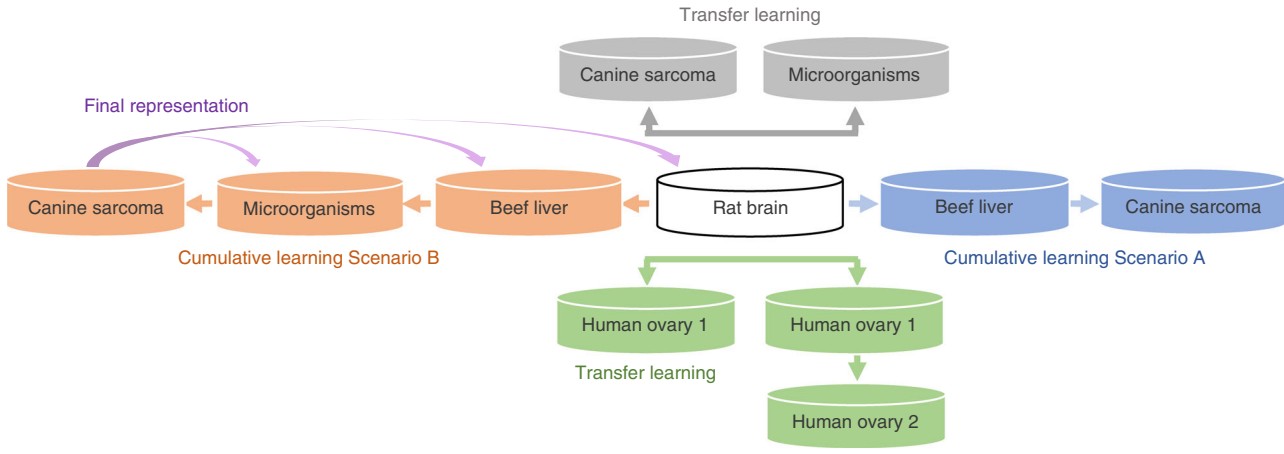

**Fig. 1 Workflow of CNNs classification: by transfer learning (in gray for canine sarcoma, microorganisms, and in green for human ovary 1).** By cumulative learning Scenario A (in blue for canine sarcoma and in green for human ovary 2). By cumulative learning Scenario B (in orange for canine sarcoma). Final representation of Scenario B is tested on the datasets used during the training (in purple arrows).

---

**Table 4 Overall accuracies of variant_LeNet architecture at classifying ovarian spectra.**

| Dataset | # classes | CNN from scratch | Transfer learning | Cumulative learning |
|---|---|---|---|---|
| Human ovary 1 | 2 | 0.78 ± 0.02 | **0.98 ± 0.00** (25%[a]) | – |
| Human ovary 2 | 2 | 0.80 ± 0.00 | 0.83 ± 0.02 (03%[a]) | **0.99 ± 0.00** (23%[a] 19%[b]) |

The best result for each task (accuracy ± standard variation over 10 independent iterations) is indicated in boldface.
[a]The improvement is expressed as a percentage relative to learning from scratch.
[b]The improvement is expressed as a percentage relative to transfer learning.

---

**Table 5 Overall accuracies of raw and preprocessed clinical spectra classification by SVM, RF, and LDA.**

| Datasets | # classes | Applied to raw datasets | | | | Applied to preprocessed datasets | | |
|---|---|---|---|---|---|---|---|---|
| | | Best CNNs | SVM | RF | LDA | SVM | RF | LDA |
| Canine sarcoma | 2 | 0.98 ± 0.00[a] | 0.77 ± 0.02 | **0.96 ± 0.01** | 0.71 ± 0.02 | 0.76 ± 0.16 | **0.96 ± 0.01** | 0.93 ± 0.02 |
| | 12 | 0.99 ± 0.00[b] | 0.61 ± 0.00 | **0.65 ± 0.04** | 0.41 ± 0.01 | 0.52 ± 0.19 | **0.65 ± 0.01** | 0.60 ± 0.04 |
| Microorganisms | 3 | 0.99 ± 0.00[c] | 0.45 ± 0.03 | 0.77 ± 0.03 | **0.90 ± 0.01** | 0.87 ± 0.02 | **0.95 ± 0.02** | 0.88 ± 0.02 |
| | 5 | 0.99 ± 0.00[c] | 0.54 ± 0.35 | **0.86 ± 0.01** | 0.67 ± 0.13 | 0.19 ± 0.09 | **0.87 ± 0.02** | 0.85 ± 0.03 |
| Human ovary 1 | 2 | 0.98 ± 0.00[c] | 0.53 ± 0.04 | **0.84 ± 0.05** | 0.65 ± 0.04 | 0.66 ± 0.24 | 0.91 ± 0.02 | **0.93 ± 0.02** |
| Human ovary 2 | 2 | 0.99 ± 0.00[b] | 0.60 ± 0.06 | **0.81 ± 0.01** | 0.71 ± 0.03 | 0.60 ± 0.05 | 0.88 ± 0.03 | **0.96 ± 0.00** |

The best result for each task (accuracy ± standard variation over 10 independent iterations) is indicated in boldface.
[a]The best CNNs from scratch.
[b]The best CNNs after cumulative learning.
[c]The best CNNs after transfer learning.

---

Transfer learning improves classification accuracy from 0.78 for training from scratch to 0.98 for the high-resolution dataset as shown in Table 4. With the low-resolution dataset, accuracy is improved from 0.80 to 0.83 by transfer learning and up to 0.99 by cumulative learning. Our CNN representation model allows a very high classification accuracy without the need for spectral preprocessing steps in contrast with the previously reported lack of sensitivity and specificity of low-resolution MS datasets.

**Comparison of our 1D-CNNs models against ML approaches.** The performance of the best 1D-CNN, that is, of all models and approaches combined, from scratch for binary canine sarcoma classification, from transfer learning for microorganisms and human ovary 1 classification, from cumulative learning for human ovary 2 (Scenario A), and for multi-class canine sarcoma (Scenario B) are compared to SVM, RF, and LDA. The comparisons are made both on raw data without feature selection and on data that has gone through preprocessing followed by a feature selection step.

The experimental results in Table 5 show that ML classification algorithms perform poorly on raw datasets as it is expected. Our CNNs models outperform the three conventional ML models (SVM, RF, and LDA) on all six datasets in terms of accuracy. The second best performance is from RF model for 5 of the 6 datasets and is from LDA for 3-class microorganism dataset. For the preprocessed datasets experiment, spectra are corrected using a sequential preprocessing of five steps as described in the "Experimental design" section. As shown in Table 5, RF gives the best result and outperforms the two other methods in SpiderMass datasets classification, especially in canine binary task, where the classification is the more accurate. LDA outperforms the two other methods for the human ovary 1 and human ovary 2 datasets classification. The computing time of

conventional algorithms is much higher than CNNs, first because the Scikit-learn library does not support GPU-based computing, and also because a few more minutes are needed to carry out the necessary preprocessing steps. Note that we test only one subset of all possible preprocessing methods that exist. In addition, some inter-spectra steps such as normalization and alignment are applied only when the signal acquisition process is completed. While our CNN models can be applied even during spectra acquisition since each test spectra could be analyzed separately.

Overall, almost all of the standard classification models, specially LDA, see notable accuracy improvement when preprocessing and feature selection methods are used. Our results show that our end-to-end CNN models outperform standard ML classification methods whether preprocessing followed by feature selection is used or not.

## Discussion
CNNs have become common tools in several research areas. They are designed to extract spatial features from input signals with different levels of abstraction. Many challenges remain fully exploiting CNNs on biomedical data, owing to data high-dimensionality, heterogeneity, and irregularity. Following their success in computer vision, the first results of deep learning methods applied to clinical data are obtained on clinical imaging (e.g. classification, segmentation, etc.). Medical images are different from ImageNet object scenes, persons, and plants, among others. Nevertheless[38–41] demonstrated that we could classify and predict outcomes from medical images using a CNN model trained on ImageNet. Authors show that the features extracted from the ImageNet database are generalizable and can be applied to alternative tasks and datasets. Our paper is inspired by these efforts on transfer learning, transferring the representation learnt on one dataset to another that intuitively do not seem to share common features, but goes beyond by accumulating the knowledge of MS data space through learning a representation on several datasets, more than two and more diverse too. We investigate here the performance of CNNs in the classification of 1D mass spectra generated for a variety of classification purposes. This study shows the use of cumulative learning for spectrum classification as a solution when the number of available samples is small. Using MALDI-MSI and other types of datasets generated in vastly different biological contexts, on different organisms, acquired by a variety of instruments, with a variety of MS ionization technologies, in different mass ranges at different resolutions, and with or without a chromatography phase. In addition, infection diagnosis can be conducted on a limited amount of cells without the need for the time-consuming bacterial culturing.

It is well known that the success of CNNs is strongly dependent on the amount of available data for its training. To overcome the limitations inherent in small numbers of training samples, we test dataset augmentation (as detailed in the Supplementary 1D-MS Data augmentation section). To the best of our knowledge, biological 1D data augmentation has not been described elsewhere, and this may be because it is not sufficient to reproduce technical variability by adding noise, baseline, and peak misalignment. Biological variability (difference between individuals) must also be introduced, in the form of relevant peak presence/absence and intensity changes. All classification accuracies on augmented SpiderMass data are below 60% (results not shown). This indicates that better understanding of biological variability is still required in order to deal with data augmentation and increase the number of samples without compromising biological information.

Preprocessing is an integral part of the multivariate analysis. However, no consensus exists on how to find the optimal preprocessing pipeline. It can be time-consuming owing to the large number of available methods and difficult to choose the best order for these methods. Since one experimenter can use different measurement devices and different measuring processes, it becomes quickly difficult to define an efficient preprocessing pipeline. For instance, normalization is based on the assumption that there are comparable numbers of ions in each spectrum, which is true in homogeneous samples, where only a few individual peak intensities change. However, in a sample, different tissues or organs may be present and express a heterogeneous set of entities having therefore different ion distributions. In such cases, normalization may not be relevant. Alignment is also a critical step for the subsequent analysis. It may be dangerous since it deforms the peak shapes (shifting, over-stretching, over-compression, etc.) to maximize the correlation between the spectra. Another limitation of the preprocessing pipeline is the step of peak picking. This step may result in a different total number of data points from one spectrum to another. It is well-known that biological information is better conserved with less manipulation, especially when dealing with clinical data (e.g. diagnosis, biomarker identification, etc.). Aligning spectra per class could positively have affected the ML results and that these results using a different alignment procedure could result in lower accuracies for the ML methods. This however will not affect the conclusions.

Our CNN models are able to classify raw MS data without the need for preprocessing steps, thus bypassing the preprocessing expertise. This performance capability is due to convolutional filters that allow CNN architecture to learn spatial peak patterns rather than only considering each $m/z$ intensity value separately as do conventional ML algorithms. More importantly, significant variations of the overall signal intensity due to biological heterogeneity and non-reproducible technical factors can be filtered by CNNs to increase the robustness of molecular pattern recognition. Combined Max-Pooling and convolution filters allowed the model to handle peak misalignment. Such end-to-end trainable systems that work with raw data offer a superior alternative to pipelines in which each step is trained independently or handcrafted to find the best combination of parameters. Inference from pre-trained models will be fast. Using raw MS data has the potential to contribute significantly to the development of a diagnosis workflow for rapid, efficient, and reliable detection of cancers or infections. Our results provide evidence that cumulative learning offers practical means of analyzing mass spectra obtained in real-world settings where the size of the dataset available for training a classification algorithm is limited. Our cumulative optimization of CNN models appeared to be better adapted than conventional ML models for mass spectra classification, even when the tasks required analysis of heterogeneous, low-resolution datasets containing several classes. In addition, cumulative CNNs appeared to offer a unified solution for classification regardless of day-to-day, sample-to-sample, and machine-to-machine variance. All this without having to worry about the quantitative assessment of the spectral quality or manual inspection, allowing thus data exchange with collaborators or from different platforms. Although we have focused our study on mass spectra, we believe that the method should be applicable to other types of analyses such as Raman spectroscopy or nuclear magnetic resonance classification tasks.

In the present study, we have investigated the performance of our learning approach for MS data classification. It would be interesting to extend the investigation to analyze which data characteristics were transferred between datasets. The focus of our future research will be the interpretation of classification models in order to identify regions of interest in spectra. Successful transfer learning between lipids and proteins species

suggests that CNNs have a more complex functioning than the simple identification of specific-phenotype peaks. Discriminating regions during the CNN classification may be explained by a complex spatial pattern recognition and by the ability of the model to generalize from one classification task to others even slightly unrelated. It would be interesting to understand how such markers influence the results and to study their global dynamic of expression.

## Methods

**Ethics approval.** Adult Wistar male rats (225–250 g, 7–8 weeks old) were sacrificed according with European and french guidelines for animal research (European Convention for the Protection of Vertebrate Animals used for Experimental and other Scientific Purposes, ETS No.123) and approved by the local Animal Ethics committee (C2EA-075 Nord-Pas de Calais). Rats were maintained and housed under pathogen-free conditions at the University of Lille Animal Care Facility.

**Materials.** These independent MS datasets are used to evaluate our proposed approaches :

1. The canine sarcoma dataset contains 1 healthy and 11 sarcoma histology types obtained from 33 annotated ex vivo biopsies[42]. Spectra are acquired in sensitivity positive ion mode using a Synapt G2-S Q-TOF instrument (Waters, Wilmslow, UK, MassLynx V4.1.SCN965). The multi-classification model presented is focused on sarcoma types only. Tumor types grading is beyond the scope of this study. No samples are excluded.

2. The microorganism dataset contains a five human pathogen collection of 2 Gram-negative bacteria, 2 Gram-positive bacteria, and 1 Yeast[43]. Spectra are acquired in negative high-resolution mode using a Synapt G2-S Q-TOF instrument (Waters, Wilmslow, UK, MassLynx V4.1.SCN965). No samples are excluded.

These two small SpiderMass datasets are characteristic of the clinical field, where samples availability coming from patients can be limited thus making the task of classification models more difficult. SpiderMass is a system designed for mobile in vivo and real-time analysis[44]. It does not involve a chromatography phase, making it compatible with rapid analysis but increasing the output spectra heterogeneity. For SpiderMass, the irradiation time is set at 10 s at 10 Hz, giving an average of 10 individual spectra (1 per laser pulse) over the 10 s period. The spectra of each microbial class are obtained from a single acquisition (2 irradiation sequences) per colony. For canine sarcoma, 7 classes are obtained from 5 irradiation sequences on one biopsy and 5 are obtained from 5 irradiation sequences on each of the different biopsies gathered from dogs of different breeds and ages.

3. The MALDI-MSI rat brain dataset contains spectra of rat gray and white brain matter, acquired using a Rapiflex MALDI-TOF instrument (Bruker Flex Imaging 5.0, Bremen). MALDI-MSI MS are imported into the user-friendly Scils software (Scils 2019a Core version) and ROI non-processed spectra are exported into a csv file format. This dataset is generated for this study. The Wistar rat (*Ratus norvegicus*) are sacrificed after behavioral practical in the University of Lille, their brains are collected and snap frozen in liquid nitrogen and stored at −80 °C. There are then sectioned at 12 μm using a cryostat (Leica microsystems) and thaw-mounted on indium thin oxide slides (LaserBio Labs). The 2,5-dihydroxybenzoic acid (DHB) matrix (Sigma-Aldrich) is sublimed onto the tissue section at 150 °C for 12 min using a 'home-built' sublimation device. The image is acquired at 50 μm × 50 μm spatial resolution in positive ion reflectron mode.

4. The beef liver dataset contains two types of spectra of liver samples from healthy animals, one acquired in positive ion mode and the other in negative ion mode, both in sensitivity mode using a Synapt G2-S Q-TOF instrument (Waters, Wilmslow, UK, MassLynx V4.1.SCN965). This dataset is generated for this study. Raw commercial product is sliced to suitable thickness, snap-frozen in liquid nitrogen and stored at −80 °C. Tissue is warmed to RT prior to SpiderMass spectral acquisition. The dataset is generated while running a time-course reproducibility experiment.

These large and medium datasets are used to investigate the transfer and cumulative learning approaches. SpiderMass is based on the WALDI process which corresponds to MALDI with water as matrix[45]. We first train the CNNs with the MALDI dataset (rat brain dataset) and then perform the classification of canine sarcoma and microorganism datasets based on this training.

5. The human ovary dataset 1 represents two classes of serums, healthy and cancerous. High-resolution spectra are acquired via ProteinChip weak-cation-exchange interaction chips (WCX2, Ciphergen Biosystems, Inc., Fremont, CA, USA) and surface-enhanced laser-desorption/ionization

(SELDI) TOF technology (QSTAR Pulsar I, Applied Biosystems, Inc., Framingham, USA)[37]. No samples are excluded.

6. The human ovary dataset 2 (as above) contains spectra acquired in low-resolution using a WCX2 protein chip via a Protein Biological System II (PBSII) SELDI-TOF instrument[37]. No samples are excluded.

We validate our transfer and cumulative learning approach using these two small well-controlled, independent, and publicly available clinical MS datasets. These two datasets are based on the SELDI process, we first train the CNNs with the MALDI dataset (rat brain dataset) and then perform ovarian classification based on this training. SELDI is an old ionization method which has small utility nowadays since it yields only a subset of the most abundant peptides and protein fragments. Some SELDI platforms may not be suitable for routine clinical diagnosis and struggle to prove their worth as reliable tools[46]. Low-resolution can make close species in *m/z* difficult to distinguish and give rise to coalesced features. Nevertheless, due to its easy-to-use quantitative screening procedures, SELDI still can be used for general description of proteins[47]. Rather than assessing the utility of these technologies or instruments, our goal in this paper is to see the problem from the user standpoint. We illustrate the strength of our methodology as a solution to multiple real-life constraints such as the fact that the user is confronted with several types of data generated by different devices or platforms, and often in a limited size.

All classes considered in this study are non-overlapping. Table 6 lists the instruments and the samples used in this study.

A simple and popular method of creating an intensity matrix from multiple spectra prior to classification is spectral bucketing or binning[48]. Easy to use on MS data, binning consists of projecting spectra into "buckets" having a fixed size. SpiderMass spectra are binned to 0.1 Da for the subsequent analyses. This binning condenses canine sarcoma and beef liver data points to 15,000 features. For the microorganism dataset it condenses points to 19,000 features for the transfer learning and to 15,000 features for the cumulative learning. We adopt a binning at 0.1 Da in this study since this window size is considered as the common size in Q-TOF MS data analysis[49,50]. Rat brain MALDI spectra are binned to have the same dimensions as SpiderMass datasets in order to allow transfer learning since CNNs require data of equal dimensions. To allow valid comparison with the original paper, both public ovarian datasets high-resolution and low-resolution are binned to *m/z* 7084[37]. Rat brain MALDI spectra are binned in this case at the same dimension as detailed in Supplementary Table 1.

We compute two of the most popular metrics to assess mass spectra similarity, namely Pearson's and cosine correlation coefficient between binned intensity spectra of the microorganism (19,000 feature vectors) and the canine sarcoma (15,000 feature vectors) datasets[51,52]. The aim is to assess whether or not spectra are correlated when they are acquired from a single physical sample. Correlation coefficients in the microorganism dataset are low between intra-colony spectra. We also find that correlation values are low between intra and inter-biopsies canine sarcoma spectra. This variability is due to the tumor heterogeneity. It is well-known that cancer is a complex and dynamic disease, where tumors are made of evolving and heterogeneous populations of cells which arise from successive appearance and expansion of subclonal populations[53]. Canine sarcoma spectra are splitted into training, validation, and test sets without taking into account their source biopsy since they are weakly correlated. The only case where spectra are highly correlated is the healthy canine samples. However the correlation coefficients are high between intra and inter-normal biopsies which means that the normal class is a good control and no matter how the normal spectra form the same biopsy are splitted into the three sets, we are not favoring a positive classification outcome. Pearson and cosine correlation coefficients of these datasets are provided in the supplementary material (Supplementary Data 1–13). The effect of taking into account the biopsy origin of the samples in the splitting of train, validation, and test samples is tested and discussed using the four canine sarcoma classes containing at least three biopsies. No significant different effect is observed (supplementary Canine spectra correlations section and Supplementary Table 54).

**Experimental design.** We import all MS datasets without undergoing any pre-processing step nor feature selection step. We bin each dataset and scale it linearly between 0 and 1. Datasets are divided randomly into three subsets, one for training, one for validation, and one for testing with ratios of 60%, 20%, and 20%, respectively. These subsets are computed via a 5-fold cross validation (CV). Classification accuracy is averaged over 10 independent iterations. These subsets are computed for each iteration using a stratified sampling to maintain the original proportion of minority classes. The loss function is weighted during the training process for samples from under-represented classes in the datasets. Performance of classifiers is measured by four metrics: global accuracy (over all classes), sensitivity, specificity, and confusion matrix as an indicator on how is simple or hard for the classifier to distinguish between different classes. CNN weights are initialized with He normal distribution since Relu/Leaky Relu is used as the activation function[54], except for the output layer, where a sigmoid function is used for binary classification or a softmax function for multi-class classification. Only the best hyper-parameters are used for the evaluation process. The model is saved only if there is an accuracy improvement of the validation set and thereby use those weights for testing.

**Table 6 Description of datasets.**

| MS instruments | Datasets | Classes | # spectra | # samples | Mass ranges | # features |
|---|---|---|---|---|---|---|
| Target domain data | | | | | | |
| Synapt G2-S Q-TOF (Waters, SpiderMass) | Canine sarcoma | Healthy | 482 | 8 | 100–1600 Da | 15,000 |
| | | Myxosarcoma | 60 | 1 | | |
| | | Fibrosarcoma | 404 | 6 | | |
| | | Hemangiopericytoma | 134 | 2 | | |
| | | Malignant peripheral nerve tumor | 60 | 1 | | |
| | | Osteosarcoma | 339 | 5 | | |
| | | Undifferentiated pleomorphic sarcoma | 376 | 5 | | |
| | | Rhabdomyosarcoma | 66 | 1 | | |
| | | Splenic fibrohistiocytic nodules | 63 | 1 | | |
| | | Histiocytic sarcoma | 105 | 1 | | |
| | | Soft tissue sarcoma | 69 | 1 | | |
| | | Gastrointestinal stromal sarcoma | 70 | 1 | | |
| | | Total | 2228 | 33 biopsies | | |
| Synapt G2-S Q-TOF (Waters, SpiderMass) | Microorganisms | *Staphylococcus aureus* | 26 | 1 | 100–2000 Da | 19,000[a] |
| | | *E. coli* D31 | 26 | 1 | | 15,000[b] |
| | | *Pseudomonas aeruginosa* | 24 | 1 | | |
| | | *Enterococcus faecalis* | 18 | 1 | | |
| | | *Candida albicans* | 23 | 1 | | |
| | | Total | 117 | 5 colonies | | |
| PBSII SELDI-TOF | Human ovary 2 | Healthy | 91 | | 700–12,000 Da | 7084 |
| | | Cancer | 162 | 37 | | |
| | | Total | 253 | | | |
| Source domain data | | | | | | |
| Rapiflex MALDI-TOF (Bruker) | Rat brain | Gray matter | 4635 | A single section | 300–1300 Da | 19,000[a] |
| | | White matter | 5465 | | | 15,000[b] |
| | | Total | 10100 | | | 7084[c] |
| Synapt G2-S Q-TOF (Waters, SpiderMass) | Beef liver | Positive mode | 1372 | 10 | 100–1600 Da | 15,000 |
| | | Negative mode | 1265 | 10 | | |
| | | Total | 2637 | 20 samples | | |
| Hybrid Quadrupole (QSTAR pulsar I) | Human ovary 1 | Healthy | 95 | | 1–20,000 Da | 7084 |
| | | Cancer | 121 | 37 | | |
| | | Total | 216 | | | |

[a]Number of features used for microorganisms transfer learning.
[b]Number of features used for canine sarcoma transfer and cumulative learning.
[c]Number of features used for ovarian transfer and cumulative learning.

*Protocol for evaluating prominent 2D-CNN adapted to 1D input.* The aim of the first experiment is to evaluate and compare the application of three prominent CNN architectures for classifying spectra in clinical datasets. We test variant_Lecun (model 1), Variant_LeNet (model 2), and Variant_VGG9 (model 3) models as detailed in the supplementary Source model optimization section. CNN architectures share the same characteristics and follow the same principles whether they are 1D or 2D. The basic difference is the dimension of the input signal and consequently how filters slide across the data. Models 1 and 3 have been described in the literature and are modified slightly to fit our 1D data classification problem. Convolutional modules and pooling size are adapted to 1D input. The same number of filters is used but they are expanded to account for spectral features larger than those extracted from images. No zero padding is needed because all of the spectra start and end with a zero value and have the same length through binning. For model 1, two fully connected layers out of three from the original LeNet architecture are kept. The adaptation of 2D-CNN architecture to 1D input was described, for example in Inception modules[34] according to data specificity. Using this approach, we expect to determine what model depth and parameters are optimal for MS spectra classification. This evaluation allows assessment of layers number required for spectral feature extraction, especially in the case of highly heterogeneous biological classes such as canine sarcoma types and in the case of low resolution such as human ovary 2 dataset.

*Protocol for evaluating transfer learning.* The aim of the second experiment is to evaluate model improvement by CNNs spectral transfer learning. The three CNN architectures are trained on the large MALDI-MSI rat brain dataset with all weights initialized according to He normal distribution. Rat brain dataset is chosen as the source domain as it contains the largest amount of spectra in our study. The

decision layers (fully connected layers and sigmoid layer) of the CNN networks are not useful, since the MALDI-MSI and clinical datasets are from different domains. The representation model weights (i.e. the convolutional portion) are then frozen so that they would not be updated during back-propagation, the decision layers are removed, and the new specific decision layers dedicated to smaller clinical datasets are trained. The evaluation of transfer learning using the canine sarcoma and microorganism datasets is illustrated in the gray portion of Fig. 1 and in panel a of Fig. 2. The green portion of Fig. 1 and the panel a of Fig. 2 illustrate the application of transfer learning to the public human ovary 1 dataset following the same strategy.

*Protocol for evaluating cumulative learning.* Transfer learning in some cases may not be enough as an aid in classifying biologically similar materials using CNN models. This proximity is reflected in a high degree of confusion between classes. This is typically the case when the biggest dataset which is supposed to be used to learn the pivotal data representation is not big enough. In addition, low-resolution or data heterogeneity can further complicate the classification task. We therefore propose two approaches to develop 1D-CNN cumulative learning:

Scenario A. The first step is to train CNN architectures on the MALDI-MSI rat brain dataset as described before for transfer learning. The representation model weights are then fine-tuned, the decision layers (i.e. fully connected and sigmoid) are removed, and new decision layers are trained with the beef liver dataset. Beef liver CNN weights (i.e. data representation) are thus initialized from the rat-brain-trained CNN representations. Finally, the beef liver CNN representation weights are frozen and new specific decision layers (fully connected and softmax) are added and trained using the canine sarcoma dataset, as illustrated in the blue portion of Fig. 1 and in panel b of Fig. 2. The green portion of Fig. 1 and the panel b of Fig. 2

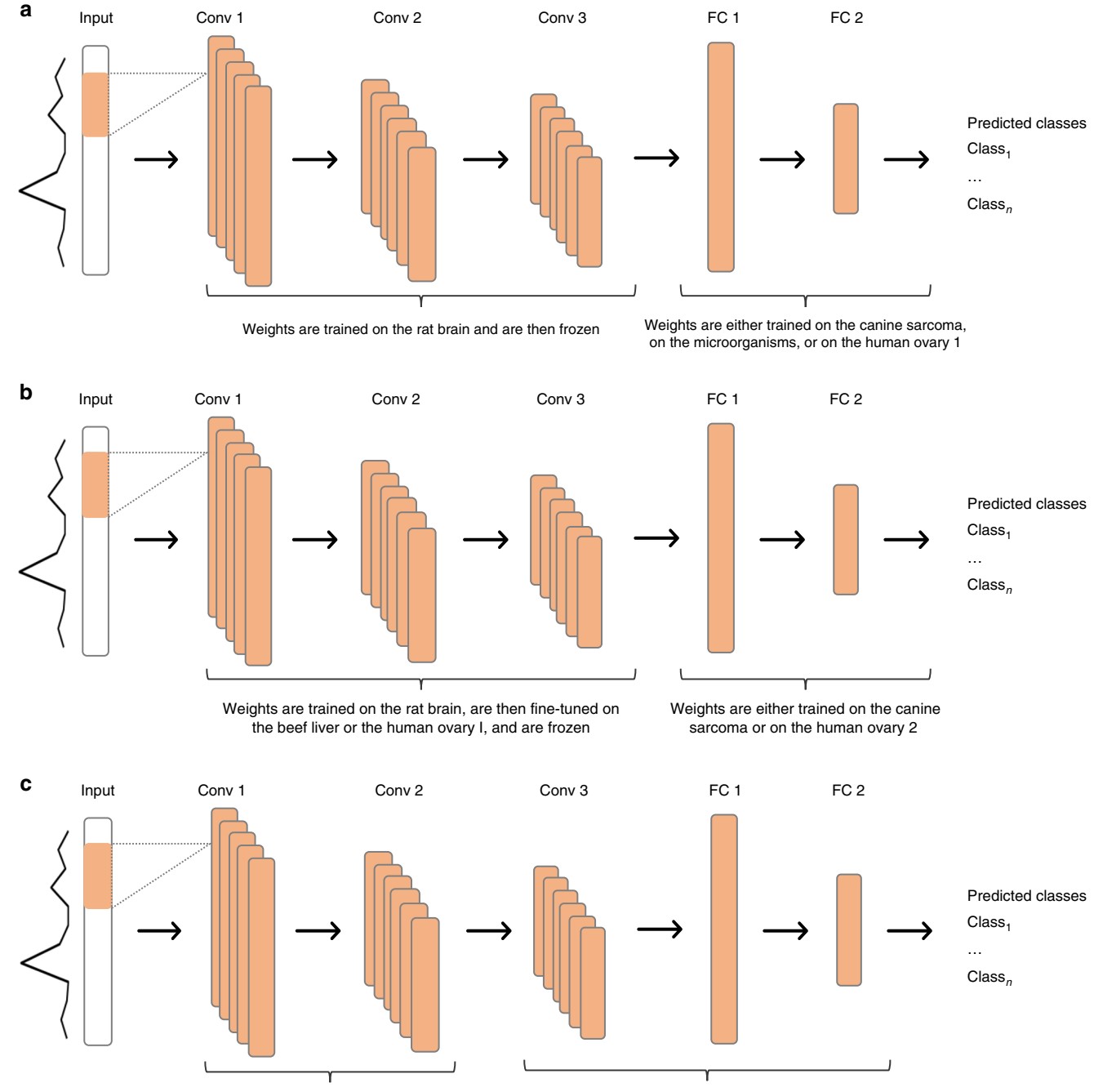

**Fig. 2 Protocols of classification with variant_LeNet architecture. a** Protocol of transfer learning. **b** Protocol of cumulative learning Scenario A. **c** Protocol of cumulative learning Scenario B.

illustrate the application of cumulative learning Scenario A to the public human ovary 2 dataset following the same strategy.

Scenario B. CNN architectures are trained on MALDI-MSI rat brain and fine-tuned with the beef liver dataset as described in Scenario A, but instead of testing this representation on the canine sarcoma dataset, an additional cumulative learning is added. Beef liver CNN representation weights are fine-tuned, decision layers (fully connected and sigmoid) are removed and new specific decision layers are added and trained using the microorganism dataset, before freezing convolutional layer weighting barring the last one and training new specific last convolutional and decision layers on the canine sarcoma dataset, as illustrated in the orange portion of Figure 1 and in panel c of Figure 2.

*Protocol for comparing our approach with other ML approaches.* Some 1D-CNNs have been found superior to conventional and popular algorithms for classifying raw data[25,33,34]. The aim of our third experiment is to compare our 1D-CNNs to three popular conventional ML algorithms, namely SVM, RF, and LDA. To make such a comparison valid, all spectra are binned similarly, and the same ratio of training, validation and test subsets is conserved. To allow a fair comparison to that of CNNs on raw data, we first evaluate the performance of these conventional ML algorithms on raw datasets. However, these conventional algorithms are not designed to classify MS spectra that have not been preprocessed. Hence, we also evaluate their performance on preprocessed datasets. In this case, spectra are corrected using sequential preprocessing means provided in the MALDIquant package (version 1.19.3)[55]. The preprocessing comprises five steps, each of these feasible using any of several methods. For the present purpose, the most standard methods are chosen: (1) log-intensity transformation. (2) Baseline subtraction using the SNIP algorithm (statistics-sensitive non-linear iterative peak-clipping). (3) Normalization by dividing mass spectra by their total ion count (TIC). (4) Alignment using a cubic warping function[16]. We align spectra by class, namely spectra of each class are aligned separately. A non-linear cubic warping function is

computed to align the whole spectra by fitting a local regression to the matched reference peaks. The cubic warping function $w(x)$ is expressed :

$$w(x) = \alpha_0 + \alpha_1 x + \alpha_2 x^2 + \alpha_3 x^3. \tag{1}$$

To estimate the model parameters ($\alpha_0... \alpha_3$), a weighted least squares is applied. (5) Peaks are detected using the median absolute deviation. Spectra are aligned prior to peak detection in order to preserve all peak information (height, width, and spatial distribution) and thereby ensure the best alignment. Ovarian datasets are preprocessed following the same preprocessing strategy. The optimized hyper-parameters for each ML algorithm are tuned with a grid search and are described in the supplementary Table 25. Only the optimal hyper-parameters are used for the evaluation. Chi-square ($\chi^2$) statistic is used to reduce data dimensionality before feeding to SVM and RF algorithms. Principal component analysis (PCA) is combined to LDA classification. We will elaborate later in the discussion section the questions that the preprocessing raises specially the undesirable effect it may have when applied to clinical data.

## Data availability
Canine sarcoma raw library is accessible on the ProteomeXchange consortium: PXD010990. Human ovarian datasets can be accessed through FDA-NCI Clinical Proteomics at https://home.ccr.cancer.gov/ncifdaproteomics/ppatterns.asp. Microorganisms, beef liver, and rat brain raw libraries are accessible on https://data.mendeley.com/datasets/33cbb37cs2/2.

## Code availability
The code that supports the findings in this study is available https://github.com/KhawlaSeddiki/1D-MS_CumulativeLearningCNNs.

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

## Acknowledgements

The operation Graham and Cedar supercomputer is funded by the Canada Foundation for Innovation (CFI), the ministére de l'Économie, de la science et de l'innovation du Québec (MESI) and the Fonds de recherche du Québec - Nature et technologies (FRQNT). This research is supported by funding from Ministere de l'Enseignement Supérieur, de la Recherche et de l'Innovation (MESRI), Institut National de la Santé et de la Recherche Médicale (Inserm), Agence Nationale de la Recherche (ANR), SATT Nord, Institut Nationale du Cancer (INCA), and Université de Lille, with the support of "Service de coopération et d'action culturelle du Consulat général de France á Québec".

## Author contributions

K.S. proposed, conducted literature analysis, designed, and conducted all the experiments. K.S., F.P., and A.D. worked on designing the methodology, analyzed the results, written the paper, and finalized the article for submission. P.S. generated the SpiderMass data. N.O. provided the MALDI-MSI data and helped to finalize the article for submission. N.O. and M.W. participated in the interpretation of the results. A.D., I.F., and M.S. obtained funds for the project. I.F. and A.D. supervised the study. All authors approved the final version of the manuscript.

## Competing interests

The authors declare no competing interests.
