## [Peer Review File · Nature Communications]

Reviewers' Comments:

Reviewer #1:

Remarks to the Author:

The authors describe the application of transfer learning and representation learning on 1D MS spectra data. By learning a CNN on 1 data set of which one has enough samples to train such network properly, and using the first convolutional part of such network in another CNN for training on a target data set of which one has just a few samples, one still can obtain good classification results. The advantage of these techniques is shown using several classification data sets. The approach is appealing, claiming that no pre-processing of the MS spectra is required, but that the MS spectra can be used directly using transfer learning and/or representation learning. This approach is borrowed from applications in (2D) image classification. The nice thing of the paper is that MS data from several instrument brands/types are used, combined and tested. The results are also compared with results obtained with classical learning algorithms (RF, LDA, SVM)

General comments:

The problem I have is that the results are based on more or less arbitrary choices (network complexity, data complexity, chosen data sets). The setup of the research is only partly structured and after reading the paper I am left with questions. The message now could be interpreted as: try several networks and if it does not work out well enough (a classification accuracy below 99%), then try several varieties of our approach and several intermediate data sets, until, hopefully, the classification does not improve anymore. I would like to see a more structural approach (experimental design like) investigating and indicating a link between the complexity of the data (number of variables, peak/mass resolution, number of peaks, ...), complexity of the CNNs, approach to be used. Some indications are present in e.g. section 3.4, varying the bin size. Secondly, but linked to the previous point: can every data set with enough samples be used for transfer learning, or are there certain requirements? Now only the rat brain data set is used. A more structured approach could have investigated this, e.g. by adapting the data (varying the number of features, peak broadening, various binning approaches, ..., number of samples, ...). It seems that the number of features of the various data sets that are used for their approach need to be made equal: what is the effect of this? Can this be done with every data set?

Three network structures with increasing complexity are more or less arbitrarily chosen, without optimizing the number of layers, the number of kernels, the size of the kernels, etc., in view of the data to be classified (mass resolution, number of variables). Why are these network structures not optimised?

Fourthly: of e.g. the canine Sarcoma data set several spectra are measured/obtained from 1 biopsy (e.g. the 404 fibrosarcoma spectra are obtained from 6 biopsies). In the validation procedure this information is not taken into account. Officially, all spectra of one biopsy (from 1 dog) should be present in either the training data set, the validation data set or the testing data set as these spectra are mutually correlated. Otherwise information of the same biopsy is present in e.g. as well the training data set and the testing data set favouring a too positive classification outcome. The same probably holds for the other data sets used to show the advantage of the investigated approach.

More detailed comments:

I would like to see in table 1 the number of features and the number of physical samples per class from which the spectra are obtained.

Page 7, halfway the protocol for comparing section: the spectra are aligned per class. This is not allowed. In this way the classification is favoured by introducing additional class related information.

Page 7, protocol for comparing section: please explain "Each microbial class was based on spectra obtained from a single acquisition." and "This was not the case for canine sarcoma where we merge spectra from different biopsies.". Please explain in more detail what is done in both

situations.

Page 7: instead of removing collinear variables PCA could be applied in combination with LDA.

Figure 3: please explain in figure caption notation principles for the various layers (e.g. conv(6,21) means 6 kernels with size 21, etc.).

Page 10, 1st paragraph: if the data are binned at 1 instead of 0.1 then I would assume that the kernel size of especially the first layer need to be adapted (in view of the remark at the end of page 7 : "Kernel sizes are large enough to cover the largest peaks in the samples so that the model do not need many layers to avoid overfitting, but not so large that detailed information is lost due to smoothing effects."

Table 6: There seems to be a small error in the RF result for the Microorganism 3-class data: 9% should be 4% (99-95%).

I do not see any information regarding the trained SVM models: were various kernel tried and optimised, which kernel is used at the end, and what are the other settings?

Several references are incomplete (32, 41, 43, 54, ...), sometimes publisher of book seems to be missing.

More or less hidden in the supplementary material is that still it seems that the number of features of the various data sets used for the transfer learning or representation learning need to be made equal: this should be present in the main paper.

Reviewer #2:

Remarks to the Author:

This survey develops a transfer learning method based on CNNs to deal with classification which has 1D spectral data in medical applications. Experiments are carried on some mass spectrometry datasets and shows its performance. It has practical value. But there are still some questions:

1. I think the design of this paper is not very well. It lacks details such as equations that formularize the proposed method in a mathematical way, for example, the proposed network structure, how it applies to the input data and how to achieve transfer should be written as formulas. Flowcharts and textual description are also recommended to provide to present the proposed framework. It is recommended to add one Section after "Introduction" and move related contents into this section to detail the model;

2. Figures are too small to see clearly, make the characters a little bigger;

3. Check terms "cumulative learning", "cumulative representation learning" and "representation learning", better use the same one;

4. The accuracy of the proposed should be contained in Table 6 for comparison.

We first want to thank the reviewers for their constructive and insightful comments which helped us to improve our article. We have considered carefully their comments and provided an answer and a revised manuscript. Below we answer the individual comments.

Reviewer 1 :

General comments:

Comment 1. The problem I have is that the results are based on more or less arbitrary choices (network complexity, data complexity, chosen data sets). The setup of the research is only partly structured and after reading the paper I am left with questions. The message now could be interpreted as: try several networks and if it does not work out well enough (a classification accuracy below 99%, then try several varieties of our approach and several intermediate data sets, until, hopefully, the classification does not improve anymore. I would like to see a more structural approach (experimental design like) investigating and indicating a link between the complexity of the data (number of variables, peak/mass resolution, number of peaks, ...), complexity of the CNNs, approach to be used. Some indications are present in e.g. section 3.4, varying the bin size.

Answer : This comment is highly relevant and addresses crucial issues of deep learning approaches. Designing the optimal or the right neural architecture for a given problem is still an open question and one of the most challenging in the field. In order to better answer this first comment, and the first comment from reviewer 2 on the formalism of transfer learning, we have added the section 2 (Models):

- In the subsection 2.1, we provide a brief overview on some of the most recent results on relating network complexity with data complexity. These works show that the standard model complexities as Vapnick-Chervonenkis or Rademacher, are over-pessimistic. They provide an upper-bound over the complexity of the network (basically the amount of parameters as a first order approximation of the space dimensions of the expressible functions). In [41], the authors even propose a new complexity measure but succeed in demonstrating its benefit over Rademacher complexity only for small networks. What can be extracted from all these works are more heuristics to design a neural architecture than a precise recipe : over-specification, regularization, batch normalization, etc.
We then describe one of the hottest current topics in the field with Neural Architecture Search. Different techniques are explored to build up iteratively a network. At each iteration, the network in its current stage is optimized to evaluate the current solution and drive the next building iteration. These techniques are very powerful but they all require a huge amount of training data to make this iterative process converging, and a lot of computations close to brute force to explore architecture possibilities. Finally, we detail the transfer learning framework which is still the most common technique to train a new model in a new context. This requires to start from a source initial architecture. We also compare the complexity of our data with the complexity of standard image datasets to provide first intuitions of network dimensions.
- In the subsection 2.2, we go through several architecture candidates as source architectures exploiting the heuristics from the previous part. We also mention some extra architecture explorations we have conducted to explain how we have focused our attention on these candidates.

- Finally in subsection 2.3, we detail the different regularization and optimization techniques we have investigated to obtain the optimal first source model for transfer learning and cumulative learning. We have in particular moved upfront in this section the general presentation of the experimental settings from section 3.3 and left only the numerical exploration of these experimental settings in section 3.3.

The design of such architectures is acknowledged to be “a bit of an art” and transfer learning is the common strategy to initiate new models for new problems from existing knowledge on similar enough problems. Our focus in this article is to show the potential of using deep learning when one has a lot of small datasets of the same kind of inputs using our brand new procedure.

Comment 2. Secondly, but linked to the previous point: can every data set with enough samples be used for transfer learning, or are their certain requirements? Now only the rat brain data set is used. A more structured approach could have investigated this, e.g. by adapting the data (varying the number of features, peak broadening, various binning approaches, number of samples, ...).

Answer : We thank the reviewer 1 for providing this detailed and relevant comment. There is no special requirement on the type of the source domain dataset, any dataset could be used but the bigger the better. The notion of “enough data” varies depending on the data properties (level of resolution, number of classes, presence or not of unbalanced classes, class ambiguities, amount of samples, dimension of each sample, etc). Unless exploring different architectures on the data by many test & error iterations, it is hard to determine in advance what is the lowest number of spectra one should have or if the number of available samples is enough to get a performing model. We detail below some elements regarding the reviewer question on the data adaptation:

1- Varying the binning size : The choice of the binning size depends on the nature of the mass analyzer and the information obtained. To allow a fair comparison with the original paper, public ovarian datasets (high-resolution and low-resolution) were binned as described in the original paper regarding the MS instrument specificities (DOI: <https://doi.org/10.1677/erc.0.0110163>). For the SpiderMass datasets, binning at 0.01 has not been tested because it was very expensive in terms of the needed computing resources (71 millions ~ 24 billions parameters) as shown in Table 1.

Table 1: Number of the three CNN architectures parameters to estimate for canine sarcoma and microorganism datasets.

CNN models	Datasets	variant_Lecun	variant_LeNet	variant_VGG9
Bin size = 1	Canine sarcoma	717.557	23.615.874	199.047.362
	Microorganisms	909.557	30.169.474	251.476.162
Bin size = 0.1	Canine sarcoma	7.197.557	244.865.410	1.969.043.650
	Microorganisms	9.117.557	310.401.410	2.493.331.650
Bin size = 0.01	Canine sarcoma	71.997.557	2.456.705.410	19.663.763.650
	Microorganisms	91.197.557	3.112.065.410	24.906.643.650

The results of the binning at 1 of the SpiderMass datasets : Canine sarcoma (1500 features) and Microorganisms (1900 features) were not included in the paper as this bin size was too aggressive and reduced considerably the spectra information. We would like to show the results of binning at 1 to the reviewer 1 (Table 2 to Table 6). If the reviewer thinks these results provide relevant complementary information to our previous results, we will add them along with the confusion matrix in the supplementary material section. We have followed the same training and evaluation protocol as described in the manuscript.

Comparison of the classification performance of CNN architectures:

Table 2 : Overall accuracy of classification of SpiderMass spectra using three CNN architectures. The best result is indicated in boldface.

Datasets	# classes	variant_Lecun	variant_LeNet	variant_VGG9
Canine sarcoma	2	0.98 ± 0.01	0.91 ± 0.05	0.95 ± 0.02
	12	0.66 ± 0.02	0.68 ± 0.08	0.72 ± 0.02
Microorganisms	3	0.90 ± 0.01	0.69 ± 0.02	0.52 ± 0.02
	5	0.75 ± 0.02	0.73 ± 0.09	0.31 ± 0.06

Transfer Learning:

Table 3: Overall accuracy of classification of SpiderMass spectra using three CNN architectures after transfer learning. The improvement in performance from scratch is expressed as a percentage.

Datasets	# classes	variant_Lecun	variant_LeNet	variant_VGG9
Canine sarcoma	12	0.81 ± 0.00 (18%)	0.88 ± 0.03 (22%)	0.87 ± 0.05 (17%)
Microorganisms	3	0.99 ± 0.00 (9%)	0.95 ± 0.01 (27%)	0.61 ± 0.01 (14%)
	5	0.99 ± 0.00 (24%)	0.99 ± 0.00 (26%)	0.81 ± 0.02 (61%)

Cumulative Learning:

Table 4: Overall accuracy of canine sarcoma (12 classes) classification by the three CNN architectures. The improvement in performance is expressed as a percentage relative to learning from scratch * , to transfer learning ** , and to cumulative learning Scenario A ***

Protocol	variant_Lecun	variant_LeNet	variant_VGG9
Scenario A	0.86 ± 0.02 (23%* 5%**)	0.95 ± 0.03 (28%* 7%***)	0.93 ± 0.02 (22%* 6%***)
Scenario B	0.90 ± 0.01 (26%* 10%** 4%***)	0.97 ± 0.00 (29%* 9%** 2%***)	0.98 ± 0.00 (26%* 11%** 5%***)

Comparison of our 1D-CNN against other ML approaches applied to raw datasets:

Table 5: Overall accuracies of SpiderMass spectra classification by SVM, RF, and LDA; percent inferiority to 1D CNN trained from scratch*, from transfer learning**, and from cumulative learning ***

Datasets	# classes	SVM	RF	LDA
Canine sarcoma	2	0.86 ± 0.03 (12%*)	0.98 ± 0.01 (0%*)	0.62 ± 0.02 (36%*)
	12	0.51 ± 0.07 (48%***)	0.69 ± 0.03 (29 %***)	0.40 ± 0.05 (59%***)
Microorganisms	3	0.39 ± 0.09 (60%**)	0.87 ± 0.04 (12%**)	0.84 ± 0.07 (15%**)
	5	0.67 ± 0.11 (32%**)	0.86 ± 0.03 (13%**)	0.82 ± 0.03 (17%**)

Comparison of our 1D-CNN against other ML approaches applied to preprocessed datasets:

Table 6: Overall accuracies of SpiderMass spectra classification by SVM, RF, and LDA; percent inferiority to the best CNN trained from scratch*, from transfer learning**, and from cumulative learning***

Datasets	# classes	SVM	RF	LDA
Canine sarcoma	2	0.91 ± 0.02 (7%*)	0.95 ± 0.10 (3%*)	0.91 ± 0.02 (7%*)
	12	0.53 ± 0.22 (46%***)	0.75 ± 0.21 (23%***)	0.72 ± 0.02 (26%***)
Microorganisms	3	0.88 ± 0.02 (11% **)	0.91 ± 0.00 (8%**)	0.88 ± 0.01 (11%**)
	5	0.88 ± 0.01 (11%**)	0.99 ± 0.01 (0%**)	0.85 ± 0.03 (14%**)

We have adopted a binning at 0.1 Da in our study since this window size is considered as the common size in Q-TOF MS data analysis (DOI:<https://doi.org/10.1016/j.ccell.2018.09.009>, DOI:<https://doi.org/10.1038/s41416-018-0048-3>, DOI: <https://doi.org/10.1038/s41598-019-39815-w>, DOI :10.1186/1477-5956-5-3 , among others references)

2- Varying the number of features : The binning is one of the ways to vary the number of features. The other way is to select a subset of the mass range (e.g. by extracting information from m/z 500-1000 instead of using the whole m/z 300-1300 Da). We have not considered this way for the rat brain dataset since it would result in a big loss of precious biological information.

3- Peak broadening : Once the spectra are acquired, we cannot physically perform mass peak broadening because it is a consequence/result of MS instrument effects (e.g. the surface charging effect) or tuning (changing delay extraction time or voltages for MALDI-TOF). For example, protein peaks are broadened because resolution is decreasing with increasing m/z range and isotopic distributions are getting broader. We don't normally observe this when looking at the low molecular mass range such as lipid peaks with a Q-TOF spectrometer. We draw your attention to the fact that SpiderMass datasets were acquired with a

Q-TOF instrument. With the MALDI-TOF instrument we have bit broader peaks in the rat brain dataset (this point is discussed in the answer of the general comment 3).

4-Varying the number of samples : We have used the MALDI rat brain dataset as the source domain because it contained the largest number of spectra. The rat brain dataset size is considerably small compared to the ImageNet database for instance (more than 14 million of images), used in most of the transfer learning examples in the literature. Our hypothesis, when the transfer learning from the rat brain dataset was not efficient enough as in the case of data heterogeneity (canine sarcoma dataset) and low-resolution (ovarian dataset), is that the unsatisfactory classification performance may be due to the fact that the source domain data was not large enough to learn an efficient data representation. We could have varied the number of rat brain spectra used for the transfer (e.g. 1000 spectra, 5000 spectra, etc.) but we think that if 10.100 spectra were not enough to learn an efficient data representation it was not useful to further reduce the number of spectra.

Overall, exploring the effect of the hyper-parameters models is a research project on its own as we have detailed in the answer of the previous comment. We did not investigate all the hyperparameters and data variation because of the small size of data. Our datasets are small and are not representative of the vast diversity of MS data to be able to draw general conclusions (e.g. only one low-resolution dataset so we cannot make generalizations about the data resolution, etc). We still can give some partial findings:

- Spectral hyper-parameters tuning has a significant impact on the classification performance. They differ from those optimized from images (pixel features 3x3). In addition, using larger filter sizes at the low level (21) and constraining them progressively gave the best result.

- Our datasets were too small and therefore we don't not have enough data to train a huge amount of parameters, thus increasing the model performances. For that reason, adding layers or testing too deep architectures such as VGG16 (16 layers) and VGG19 (19 layers) as it was indicated in the paper did not improve the classification results. It led to a situation where the data is too small in comparison to the huge number of parameters to estimate (Table 1).

Rather than defining new methods or new ways to approach existing strategies to optimize CNN models/hyperparameters by exploring all the possibilities and to vary data which in any case would not improve much the model performance, we have focused our efforts to develop a new classification methodology using deep learning approaches when the available data are small.

Comment 3. The reviewer pointed out that the number of features of the various data sets that are used for their approach need to be made equal. He asked what is the effect of this and if this can be done with every dataset?

Answer : As the reviewer noted, all datasets used for the transfer and cumulative learning must have the same size because the neural network has a fixed number of inputs so vectors fed to the neural network must have the same size. We have chosen the binning as a way to equalize data size. The MALDI rat brain dataset was binned in three different ways depending on whether it has been used for transfer or cumulative learning as it is shown in the column “# features” in Table 1 of the revised manuscript.

The bigger the bin size, the more information will be summarized in the resulting intensity matrix. Typically in the MALDI-TOF datasets, peak width increases with increasing m/z values (DOI:

<http://dx.doi.org/10.1016/j.bbapap.2013.01.040>). This peak broadening in the rat brain dataset requires to use larger bin size (0.1 or 0.2) to take into account this broadening, in addition to other instrument variation (e.g. masses which are repeating themselves). In the following Table 7 we have computed the classification accuracy of the rat brain dataset to assess the effect of the different binning sizes on the CNN classification performances.

Table 7: Overall accuracy of the rat brain dataset with three CNN architectures

# features	variant_Lecun	variant_LeNet	variant_VGG9
Without binning (38500 features)	0.995 ± 0.001	0.995 ± 0.001	0.994 ± 0.002
Binning at 19.000 features	0.996 ± 0.002	0.994 ± 0.002	0.997 ± 0.000
Binning at 15.000 features	0.996 ± 0.000	0.990 ± 0.001	0.994 ± 0.001
Binning at 7084 features	0.989 ± 0.002	0.990 ± 0.002	0.992 ± 0.001

As shown in the results of Table 7, the performance of the three CNNs models does not decrease significantly with the different binning size. While binning summarizes spectral information, data representation is preserved and is efficiently transferred to the other datasets. Nevertheless, binning at 7084 features requires a longer training time (30 epochs) and a reduction of the learning rate (division by 0.1 every 10 epochs) when the validation set accuracy has stopped improving. Our future research will benefit from all the advances in understanding the effect of compression on the extracted data representation.

The reviewer pointed out that the three network structures with increasing complexity are more or less arbitrarily chosen, without optimizing the number of layers, the number of kernels, the size of the kernels, etc., in view of the data to be classified (mass resolution, number of variables). He asked why are these network structures not optimised?

Answer : We have described in the Hyper-parameter optimization section (#3.3) our investigation strategy of the hyper-parameters search. This section comprises: optimal filter sizes search, a comparison of the method used to handle overfitting: Batch normalization, Dropout, $l1/l2$ regularization. The best learning rate (0.1, 0.01, and 0.001). The type of optimizer algorithms: Adam and Stochastic gradient descent (SGD), etc. Instead of choosing a single architecture and varying its number of layers and kernels, we have chosen to adapt three proven architectures where each one contains different numbers of layers (three, five, and nine layers) to extract spectral spatial patterns. We have tested simultaneously these three models where each of them contained different numbers of filters and kernel sizes. We also have tried to do data augmentation to compensate for the small number of samples, but this practice is still hard to apply on biological data as we have explained in the discussion section (# 5). Again, as we detailed in the answer to general comments 1 and 2, because of its high cost we have explored the hyperparameters in predefined ranges with a grid-search approach.

Comment 4. The reviewer pointed out the case of the canina sarcoma dataset where several spectra are measured/obtained from 1 biopsy (e.g. the 404 fibrosarcoma spectra are obtained from 6 biopsies). In the validation procedure this information is not taken into account. Officially, all spectra of one biopsy (from 1 dog) should be present in either the training data set, the validation data set or the testing data set as

these spectra are mutually correlated. Otherwise information of the same biopsy is present in e.g. as well the training data set and the testing data set favouring a too positive classification outcome. The same probably holds for the other data sets used to show the advantage of the investigated approach.

Answer : This comment is very relevant. Actually, we had already identified the question but being focused on the description of our new approach to make compliant (very) small datasets with training deep networks we omit to report it. To confirm that we could reasonably run the experiments done in the paper, we have evaluated the correlation of spectra intra- and inter-classes. Indeed it is expected that spectra resulting from the same biopsy and so the same patient/organ would be highly correlated. However, cancer is a complex, dynamic organism and during the course of the disease becomes more heterogeneous inter patient as well as intra patient. One single biopsy or excised tissue can express a diverse set of tumor cells. This is why it is important to acquire several spectra from different tissue regions in order to capture the totality of the tumor heterogeneity.

We have calculated Pearson's and cosine correlation coefficient as metrics to determine the similarity of mass spectra within a single biopsy. Pearson's correlation is probably the most common practice in mass spectrometry analysis due to its high performance in capturing spectral similarity [doi:10.1371/journal.pone.0164260, DOI https://doi.org/10.1007/978-3-319-31744-1_11, <http://hdl.handle.net/10919/36109>]. [DOI <https://doi.org/10.1186/1471-2105-12-235>, DOI: [10.1109/BIBM.2011.113](https://doi.org/10.1109/BIBM.2011.113)]. It has also been used to assess similarity in mass spectrometry data for tasks ranging from protein identification, peaks alignment, spectra clustering, to quality control [doi:10.1186/1471-2105-6-S2-S5, <https://doi-org.acces.bibl.ulaval.ca/10.1021/pr800214d>, doi: 10.3389/fmicb.2018.03000].

We have computed the Pearson's and cosine correlation coefficient between all pairs of the binned intensity spectra from all canine biopsies (15.000 feature vectors) and microorganisms (19.000 feature vectors). The excel file containing the result is attached in the supplementary material. Results show low Pearson correlation and cosine similarity values between intra-cancer canine biopsies spectra. This variability is due to tumor heterogeneity. The same applies to the microorganism dataset, where we found that Pearson and cosine correlation were low between inter and intra-colonies spectra. Correlation coefficients were high only between spectra from the same normal biopsy and from two normal biopsies which indicates that splitting the spectra by putting each biopsy either into the train, validation, or test sets or by splitting spectra randomly will not have an effect of accuracy since the majority of spectra are very highly correlated. We do not expect normal tissue to be heterogeneous.

In order to evaluate the potential correlation between spectra, we did not only evaluate correlation coefficients on the input, but also evaluate the impact on the model outcome. We had initially assessed the impact on the model of classifying spectra from the same sample after the classification step by analyzing the result of the sensitivity, and specificity, and the confusion matrix (Table 3 and Table 4 that were provided in the supplementary material of the manuscript). To better illustrate this independence of the model to the input data split, we present here each model apart and show in the Table 8 below that the three classification models appear to not favor particularly classes obtained from only one biopsy.

Table 8 : Balanced accuracy (sensitivity+specificity/2) of canine sarcoma classes. * Classes with more than 1 biopsy, ** Classes with only 1 biopsy

Healthy*	variant_lecun: 0.992 variant_leNet: 0.975
----------	--

	variant_VGG9: 0.984
Fibrosarcoma*	variant_lecun: 0.905 variant_leNet: 0.915 variant_VGG9: 0.915
Hemangiopericytoma*	variant_lecun: 0.910 variant_leNet: 0.945 variant_VGG9: 0.880
Osteosarcoma*	variant_lecun: 0.925 variant_leNet: 0.875 variant_VGG9: 0.920
Undifferentiated pleomorphic sarcoma*	variant_lecun: 0.940 variant_leNet: 0.930 variant_VGG9: 0.935
Myxosarcoma**	variant_lecun: 0.857 variant_leNet: 0.955 variant_VGG9: 0.950
Malignant peripheral nerve tumor**	variant_lecun: 0.825 variant_leNet: 0.965 variant_VGG9: 0.995
Rhabdomyosarcoma**	variant_lecun: 0.935 variant_leNet: 0.945 variant_VGG9: 0.955
Splenic fibrohistiocytic nodules**	variant_lecun: 0.880 variant_leNet: 0.930 variant_VGG9: 0.915
Histiocytic sarcoma**	variant_lecun: 0.970 variant_leNet: 0.910 variant_VGG9: 0.970
Soft tissue sarcoma**	variant_lecun: 0.900 variant_leNet: 0.905 variant_VGG9: 0.915
Gastrointestinal stromal sarcoma**	variant_lecun: 0.890 variant_leNet: 0.860 variant_VGG9: 0.875

In addition, to show that our way of splitting the spectra is not favouring a too positive classification outcome, we have computed classification accuracy for the 4 canine sarcoma classes containing at least 3 biopsies: Healthy (H, 8 biopsies) , Fibrosarcoma (F, 6 biopsies), Osteosarcoma (5 biopsies), Undifferentiated pleomorphic sarcoma (U, 5 biopsies) to assess the effect of spectra splitting on the performance of CNN models. We followed the same protocol for training models as described in the paper.

Split 1: Spectra from the same biopsy are present in either the training set, the validation set or the testing set.

Split 2 : Spectra are splitted randomly at each iteration without taking into account their biopsy.

Table 9 : Overall accuracy and confusion matrix for 4 canine sarcoma classes

	variant_lecun					variant_lecun					variant_VGG				
Split 1	0.91 ± 0.04					0.88 ± 0.03					0.91 ± 0.02				
		H	F	O	U		H	F	O	U		H	F	O	U
	H	99	0	0	0	H	98	0	0	2	H	110	0	0	4
	F	0	68	8	2	F	1	60	7	6	F	0	69	11	1
	O	0	2	46	2	O	1	4	59	10	O	0	5	54	2
	U	4	1	2	62	U	4	0	0	71	U	0	3	1	59
Split 2	0.93 ± 0.02					0.86 ± 0.02					0.90 ± 0.01				
		H	F	O	U		H	F	O	U		H	F	O	U
	H	90	0	1	0	H	74	0	0	0	H	99	0	3	8
	F	0	73	4	0	F	0	68	2	1	F	0	79	1	3
	O	0	4	65	1	O	10	13	74	13	O	0	7	57	5
	U	3	4	4	79	U	10	2	1	64	U	3	2	3	63

As shown in Table 9, classification results are very close and the split we have adopted in the paper (split 2) is not favouring a too positive classification outcome.

We included the following paragraph in the paper to explain this point: “We compute two of the most popular metrics to assess mass spectra similarity, namely Pearson’s and cosine correlation coefficient between binned intensity spectra of the microorganism (19.000 feature vectors) and the canine sarcoma (15.000 feature vectors) datasets [<https://doi.org/10.1038/s41598-018-37560-0>, <https://doi.org/10.1371/journal.pone.0164260>]. The aim is to assess whether or not spectra are correlated when they are acquired from a single physical sample. Correlation coefficients in the microorganism dataset are low between intra-colony spectra. We also find that correlation values are low between intra and inter-biopsies canine sarcoma spectra. This variability is due to the tumor heterogeneity. It is well-known that cancer is a complex and dynamic disease, where tumors are made of evolving and heterogeneous populations of cells which arise from successive appearance and expansion of subclonal populations [<https://doi.org/10.1371/journal.pone.0224143>]. Canine sarcoma spectra are splitted into

training, validation, and test sets without taking into account their source biopsy since they are weakly correlated. The only case where spectra are highly correlated is the healthy canine sarcoma samples. However the correlation coefficients are high between intra and inter-normal biopsies which means that the normal class is a good control and no matter how the normal spectra from the same biopsy are splitted into the three sets, we are not favoring a positive classification outcome. Pearson and cosine correlation coefficients of datasets are provided in the supplementary material.”

If the reviewer finds useful to add all the other verifications on the non-correlation of spectra, exposed above, and that adding them will not attenuate the focus on the core methodology, we can add them to the supplementary material.

More detailed comments:

Comment 1. The reviewer asked to see in Table 1 the number of features and the number of physical samples per class from which the spectra are obtained.

Answer : We have added three new columns into Table 1 in the Datasets subsection (# 3.2) : a column “# samples” which contains the physical number of samples. A column “mass range” which contains the mass range expressed in Da of each sample. A column “# features” which contains the number of features of each sample to provide a complete description of the used datasets.

Comment 2. The reviewer pointed out in the protocol for comparing section (page 7): the spectra are aligned per class. This is not allowed. In this way the classification is favoured by introducing additional class related information.

Answer : We thank the reviewer for pointing out this part of the preprocessing protocol and highlighting rightly the danger of potentially introducing a bias when aligning each class spectra separately. To extract meaningful information from a group of samples, it is desirable that different spectra have the same baseline, the same level of noise, and that peaks corresponding to the same biological entity show up at the same location for different spectra. In real-world experiments, spectra are obtained from different samples/experiments, have different baselines, different levels of noise, and are not aligned. (1) The baseline is a varying component of a spectrum with high values in the region of low m/z values and vanishing for high m/z values. (2) The noise level depends on the measurement device and the measuring process [DOI: [10.1016/j.bbapap.2013.01.040](https://doi.org/10.1016/j.bbapap.2013.01.040)]. (3) Peaks corresponding to the same entity do not show up at the same m/z location.

From our perspective we have several concerns about applying preprocessing to clinical data. Although the preprocessing pipeline is imperative for spectral data subsequent analysis, we believe that it can make the extraction of meaningful biological information difficult. Impose a certain uniformity to data and thus force the resemblance between the spectra (e.g. by equalizing the total ion count under the assumption that the number of ions must be equal from spectrum-to-spectrum, stretching/compression of peaks until they become highly correlated and aligned, imputation of intensity values when the number of detected peak is different, etc) runs counter to the biological variability. In addition, no consensus exists regarding the best order of different pre-processing steps and if all steps must be applied or not (e.g. apply the denoising step to the whole dataset which may contain a changeable level of noise across its spectra). Many factors such as sample matrix, equipment settings, and environmental influences can affect the

data, so that previously applied preprocessing methods may not work well on a new dataset. Especially if one is working on datasets coming from different platforms or multiple instruments, it can be hard to define a unique solution for all of them. We totally agree with the reviewer that aligning spectra from each class separately may introduce a bias in the classification step and can make the distinction easier for the classification algorithms. Even normalizing and aligning all the spectra together (merging different tissues spectra) may lead to a bias when seeking to increase the correlation between the spectra by distorting the signal during the alignment step.

In addition to the comparison between CNNs and ML approaches (SVM, RF, and LDA) applied to preprocessed datasets (# 3.4 Experimental design section), we have added in the revised manuscript a comparison between CNNs and ML approaches (SVM, RF, and LDA) applied to raw datasets (# 3.4 Experimental design section) to convince the readers of the usefulness of our CNN approaches effective without the need for data preprocessing and dimensional reduction.

We have included this paragraph in the discussion section of the revised manuscript to detail the negative effect the processing may have on biological data “Preprocessing is an integral part of the multivariate analysis. However, no consensus exists on how to find the optimal preprocessing pipeline. It can be time-consuming owing to the large number of available methods and difficult to choose the best order for these methods. Since one experimenter can use different measurement devices and different measuring processes, it becomes quickly difficult to define an efficient preprocessing pipeline. For instance, the normalization step is based on the assumption that there are comparable numbers of signals present in each spectrum, which is true in homogeneous samples, where only a few individual peak intensities change. However, in a sample, different tissue or organs may be present and express a heterogeneous set of entities having therefore different ion distributions. In such cases, normalization may not be relevant. Alignment is also a critical step for the subsequent analysis. It may be dangerous since it deforms the peak shapes (shifting, over-stretching, over-compression, etc.) to maximize the correlation between the spectra. Another limitation of the preprocessing pipeline is the step of peak picking. This step may result in a different total number of data points from one spectrum to another. It is well-known that biological information is better conserved with less manipulation, especially when dealing with clinical data (e.g. diagnosis, biomarker identification, etc.).”

Comment 3. The reviewer asks for an explanation “Each microbial class was based on spectra obtained from a single acquisition.” and “This was not the case for canine sarcoma where we merge spectra from different biopsies.”. Please explain in more detail what is done in both situations.

Answer : We thank the reviewer for bringing this point to our attention and we want to clarify this paragraph. The irradiation time of the SpiderMass is approx. 10 seconds. For the SpiderMass datasets:

- Microorganism dataset was obtained from 5 acquisition files. Each file contains 2 sequences of irradiation per colony (20 seconds).

- Canine sarcoma dataset was obtained from 165 acquisition files. Each file contains 1 sequence of irradiation per biopsy (10 seconds). Some classes were obtained from different files. For instance, the Osteosarcoma class was obtained from 5 different files and consequently from 5 biopsies, these 5 biopsies were obtained from 5 dogs of different breeds and ages. We have detailed the number of biopsies of each class in Table 1 (# 3.2 Datasets subsection). We apologize and corrected the following

mistake in the revised manuscript, acquisitions were done the same day and not spread over days as it was written in the original paper.

We have included the following sentences into the text to clarify this point: “The spectra of each microbial class are obtained from a single acquisition (2 irradiation sequences) per colony. For canine sarcoma, 7 classes are obtained from five irradiation sequences on one biopsy and 5 are obtained from 5 irradiation sequences on each of the different biopsies gathered from dogs of different breeds and ages.”

Comment 4. The reviewer suggests to use PCA combined with LDA instead of removing collinear variables.

Answer : As the reviewer requested, we have modified the analysis with a combination PCA-LDA instead of a combination Chi2-LDA. Classification results are shown in Table 7 of the manuscript (# 4.4.1 subsection).

Comment 5. The reviewer asks for an explanation of Figure 3 caption notation : for the various layers (e.g. conv(6,21) means 6 kernels with size 21, etc.).

Answer : We have modified the position of Figure 3 that became Figure 1. We have included the following explanation in its caption notation : “Conv(6,21) means 6 kernels with size 21. Pooling(2,2) means pool size of 2 with a stride of 2”.

Comment 6. Page 10, 1st paragraph: if the data are binned at 1 instead of 0.1 then I would assume that the kernel size of especially the first layer need to be adapted (in view of the remark at the end of page 7 : “Kernel sizes are large enough to cover the largest peaks in the samples so that the model do not need many layers to avoid overfitting, but not so large that detailed information is lost due to smoothing effects.”

Answer : In the flowing experiment, we define the peak width as the number of intensity values between two local minimums. To answer this comment, we have counted the average of peak widths in the canine sarcoma and rat brain datasets at different binning.

Table 10: Quantiles of peaks width for canine sarcoma and rat brain datasets at different binning

Datasets	Binning size	1st Qu.	Median	Mean	3rd Qu.
Rat brain	Without binning (38500 features)	1.0	2.0	2.1	3.0
	Binning at 15.000 features	1.0	2.0	2.3	3.0
Canine sarcoma	Bin=1 (1500 features)	1.0	2.0	2.4	3.0
	Bin=0.1 (15.000 features)	1.0	3.0	6.8	7.0

The results in Table 10 show that peak widths do not change much when the binning size changes (median column). Values are close and consequently the same network could be used for the two binning sizes (1 and 0.1) with the canine sarcoma dataset without the need to adapt the filter sizes. We choose to set the filter size of the first convolutional layer at 21 for the three architectures so that it covers on average 3 ~ 5 peaks. As we mentioned previously in the answer of the general comment 2, Q-TOF

analyzer didn't produce peak broadening and therefore has stable peak widths along the m/z axis. This filter size is a trade-off to cover the largest peaks while avoiding overfitting. In addition, as we have explained in the answer of the general comment 2, binning at 1 reduces considerably the number of the estimated parameters (approx. divided by 10 for the variant_VGG9 model) which enlighten the high performance of variant_VGG9 model on the canine sarcoma dataset.

Comment 7. The reviewer notes an error in Table 6: There seems to be a small error in the RF result for the Microorganism 3-class data: 9% should be 4% (99-95%).

Answer : We thank the reviewer for catching this error which was corrected.

Comment 8. The reviewer notes the information regarding the trained SVM models: used kernel and optimised hyper-parameters.

Answer : We detail the grid search of the optimized hyper-parameters for each ML algorithm (SVM, RF, and LDA) in the supplementary section (Table 23). We add a reference to this table into the main text using the following sentences : “The optimized hyper-parameters for each ML algorithm are tuned with a grid search and are described in Table 23 in the supplementary section”

Comment 9. The reviewer notes several incomplete or missing references (32, 41, 43, 54, ...).

Answer : We thank the review for noticing this error. We have checked all references of the revised manuscript to make sure that they are correct.

Comment 10. The reviewer notes that features information used for the transfer learning or cumulative learning are more or less hidden in the supplementary material, instead they should be present in the main paper.

Answer : We thank the review for this comment. We have improved the content of the datasets subsection (#3.2) for clarity. We have moved the paragraph explaining the application of the binning and the number of used features from the supplementary material to the main text.

Reviewer 2 :

Comment 1. The reviewer notes that the paper lacks details such as equations that formalize the proposed method in a mathematical way, for example, the proposed network structure, how it applies to the input data and how to achieve transfer should be written as formulas. He pointed out also to use flowcharts and textual description to present the proposed framework.

Answer : We have added a mathematical formulation to formalize the transfer learning approach in Section 2.1. We added a flowchart in Figure 2 to give an overview of the cumulative learning approach from multiple source domains compared to the traditional transfer learning system with a single source domain. Later in the paper, we have explained in detail our cumulative learning framework (#3.4 Experimental design) by a flowchart in Figure 4 where the colour code is specific to each learning protocol and the size of cylinder is proportional to dataset sizes.

Comment 2. The reviewer recommends to add one Section after "Introduction" and move related contents into this section to detail the model.

Answer : We have revised the respective paragraphs according to this recommendation and we have added new sections as we answered the first comment of review 1:

a- The contribution paragraph were moved to the introduction section (# 1) where we detail the novelties of our cumulative framework compared to the traditional transfer learning and the existing approaches described in the literature.

b- A “Models” section (#2) divided in three parts:

- Neural Architecture Design (# 2.1) where we discuss the strategy one can follow to design an adequate architecture for a new classification problem. Then we discuss some points related to the data and the problem complexity.
- Source model candidates (# 2.2) where we introduce the existing CNN models developed for 1D signal analysis. Then, we explain our CNN architecture choices according to our MS signal specificities.
- Source model optimization (# 2.3) where we describe the final CNN architectures hyper-parameters.

c- General overview subsection (#3.1 in the Methodology section) where we provide a general overview of our framework before detailing the evaluation protocol into the Experimental design section (# 3.4).

Comment 3. The reviewer notes that figures are too small to see clearly.

Answer : We have enlarged all the figures and made the characters bigger to make them more clear.

Comment 4. The reviewer notes that it is better to use one term instead of different terms "cumulative learning", "cumulative representation learning" and "representation learning".

Answer : We have modified all the text and we kept only the term "cumulative learning" in reference to our approach.

Comment 5. The reviewer notes that the accuracy of our models should be contained in Table 6 to make the comparison easy.

Answer : To allow readers to understand more easily the comparison with the conventional ML algorithms (SVM, RF, and LDA), we have added a column with the accuracy of our best CNN models into Table 6 (conventional ML algorithms applied to raw datasets) and into Table 7 (conventional ML algorithms applied to preprocessed datasets).

Following the reviewer 1 and reviewer 2 comments, we have changed the whole structure of the paper and we added new sections.

Reviewers' Comments:

Reviewer #1:

Remarks to the Author:

I accept most answers and modifications of the authors. Several additionally obtained results can be added to the Supplemental Material, as indicated below. However, one question is not taken into account appropriately, and I suggest to modify the manuscript as indicated below (see 'Regarding the answer to detailed comment 2').

Regarding Answer to Comment 2:

I think the results of varying the binning size, ..., can be added to the Supplementary information. It provides additional useful information to the reader and shows that the authors have investigated these effects. Comparing the tables 2-6 in the rebuttal with the results of binning at 0.1 as presented in the paper, then the results are frequently similar and sometimes (slightly) worse.

The authors can also add to the Supplementary information their remark regarding variation of the number of samples and the reason why they did not investigate this (see '4-Varying the number of samples' in the rebuttal).

Regarding Answer to comment 3: the authors can also add these results to the Supplementary information as additional

Regarding Answer to Comment 4: the authors could add to the already added text in the paper that "the effect of taking into account the biopsy origin of the samples in the splitting in test and training samples was further tested using the 4 canine sarcoma classes containing at least 3 biopsies. No significant different effect was observed." And refer to table 9, added to the Supplemental material.

Regarding the answer to detailed comment 2: there are counter arguments possible against the statement of the authors that 'Impose a certain uniformity to data and thus force the resemblance between the spectra (e.g.) runs counter to the biological variability.'. For instance: by pre-processing one wants to correct for (unwanted) experimental variation, resulting from measurement artefacts (peak shifts, MS detector sensitivity variation, MS detector drift, etc.). Leaving this variability in the data could on the other hand by chance due to faulty analysis practise correlate with the property under investigation (such as class difference) and in that way lead to wrong or too positive results.

I understand that the authors did not adjust their data alignment procedure and still align per class. According to me and literature this is not correct, despite their lengthy answer. A way out could be by adding to the manuscript either in the discussion on page 16 or on in section 4.4.2 a statement that 'Aligning per class could positively have affected the ML results and that these results using a different alignment procedure could result in lower accuracies for the ML methods. This however will not affect the conclusions.' Such a statement will not affect the overall conclusions of the paper.

Small remark: 7th line on page 10: I assume that pic should be peak.

Reviewer #2:

None

We first want to thank the reviewer 1 for his constructive and insightful comments which helped us to further improve our article. We have considered carefully his comments and provided an answer and a revised manuscript. Below we answer the individual comments.

Reviewer 1 :

Comment 1. Regarding Answer to Comment 2: I think the results of varying the binning size, ..., can be added to the Supplementary information. It provides additional useful information to the reader and shows that the authors have investigated these effects. Comparing the tables 2-6 in the rebuttal with the results of binning at 0.1 as presented in the paper, then the results are frequently similar and sometimes (slightly) worse.

Answer : We added supplementary “CNN architectures performances with 1D-MS data binned to 1” and “ML approaches performances with 1D-MS data binned to 1” sections to the revised manuscript (supplementary Tables 50 to Table 53, p.17).

Comment 2. The authors can also add to the Supplementary information their remark regarding variation of the number of samples and the reason why they did not investigate this (see ‘4-Varying the number of samples’ in the rebuttal).

Answer : We added a supplementary “Data exploration” section (Varying the number of samples and Varying the bin size, p.6) to the revised manuscript.

Comment 3. Regarding Answer to comment 3: the authors can also add these results to the Supplementary information as additional.

Answer : We added the answer of comment 3 in the supplementary “Matrix construction” section of the revised manuscript (supplementary Table 1).

Comment 4. Regarding Answer to Comment 4: the authors could add to the already added text in the paper that “the effect of taking into account the biopsy origin of the samples in the splitting in test and training samples was further tested using the 4 canine sarcoma classes containing at least 3 biopsies. No significant different effect was observed.” And refer to table 9, added to the Supplemental material.

Answer : We added a supplementary “Canine spectra correlations” section (Supplementary Table 54, p.18) with a reference to the table in the main manuscript.

Comment 5. Regarding the answer to detailed comment 2: there are counter arguments possible

against the statement of the authors that ‘Impose a certain uniformity to data and thus force the resemblance between the spectra (e.g.) runs counter to the biological variability.’. For instance: by pre-processing one wants to correct for (unwanted) experimental variation, resulting from measurement artefacts (peak shifts, MS detector sensitivity variation, MS detector drift, etc.). Leaving this variability in the data could on the other hand by chance due to faulty analysis practise correlate with the property under investigation (such as class difference) and in that way lead to wrong or too positive results. I understand that the authors did not adjust their data alignment procedure and still align per class. According to me and literature this is not correct, despite their lengthy answer. A way out could be by adding to the manuscript either in the discussion on page 16 or on in section 4.4.2 a statement that ‘Aligning per class could positively have affected the ML results and that these results using a different alignment procedure could result in lower accuracies for the ML methods. This however will not affect the conclusions.’ Such a statement will not affect the overall conclusions of the paper.

Answer : We added the suggested remark ‘Aligning spectra per class could positively have affected the ML results and that these results using a different alignment procedure could result in lower accuracies for the ML methods. This however will not affect the conclusions’ to the “Discussion” section of the revised manuscript (p.7).

Comment 6. Small remark: 7th line on page 10: I assume that pic should be peak.

Answer : We corrected the mistake.